# Evaluation of Prebiotics through an In Vitro Gastrointestinal Digestion and Fecal Fermentation Experiment: Further Idea on the Implementation of Machine Learning Technique

**DOI:** 10.3390/foods11162490

**Published:** 2022-08-17

**Authors:** Hokyung Song, Dabin Jeon, Tatsuya Unno

**Affiliations:** 1Subtropical/Tropical Organism Gene Bank, Jeju National University, Jeju 63243, Korea; 2Faculty of Biotechnology, School of Life Sciences, Sustainable Agriculture Research Institute (SARI), Jeju National University, Jeju 63243, Korea

**Keywords:** prebiotics, probiotics, gut microbiome, high throughput sequencing, in vitro, gastrointestinal digestion, fecal fermentation, machine learning

## Abstract

Prebiotics are non-digestible food ingredients that promote the growth of beneficial gut microorganisms and foster their activities. The performance of prebiotics has often been tested in mouse models in which the gut ecology differs from that of humans. In this study, we instead performed an in vitro gastrointestinal digestion and fecal fermentation experiment to evaluate the efficiency of eight different prebiotics. Feces obtained from 11 different individuals were used to ferment digested prebiotics. The total DNA from each sample was extracted and sequenced through Illumina MiSeq for microbial community analysis. The amount of short-chain fatty acids was assessed through gas chromatography. We found links between community shifts and the increased amount of short-chain fatty acids after prebiotics treatment. The results from differential abundance analysis showed increases in beneficial gut microorganisms, such as *Bifidobacterium*, *Faeclibacterium*, and *Agathobacter*, after prebiotics treatment. We were also able to construct well-performing machine-learning models that could predict the amount of short-chain fatty acids based on the gut microbial community structure. Finally, we provide an idea for further implementation of machine-learning techniques to find customized prebiotics.

## 1. Introduction

Prebiotics are defined as “a non-digestible food ingredient that beneficially affects the host by selectively stimulating the growth and/or activity of one or a limited number of bacteria in the colon, and thus improves host health” [1]. Therefore, prebiotics are resistant to gastric acid, hydrolysis by mammalian enzymes, and gastrointestinal absorption [2]. Prebiotics stay intact in the upper gastrointestinal tract, which is one of their important properties.

The favorable gut microbes stimulated by prebiotics are predominantly short-chain fatty acid (SCFA)-producing bacteria—for example, *Bifidobacterium*, *Faecalibacterium*, and *Prevotella*—and several other species, such as *Ruminococcus bromii*, which provide substrates to (cross-feed) other species by degrading prebiotics. The main end-products of prebiotics degradation are, therefore, SCFAs, which exert numerous beneficial effects on human health [3]. They play an important role in gut homeostasis by maintaining colonic pH, decreasing the abundance of pathogenic bacteria [4], and regulating gut permeability [5]. In addition, they are small enough to diffuse into the blood stream, thereby affecting distant organs as well. Together, SCFAs act as important metabolites regulating energy metabolism and influencing endocrine and immune functions [6].

To date, diverse health-promoting foods, including fruits, vegetables, mushrooms, seaweeds, cereals, and ginseng, have been tested for their potential as prebiotics themselves or as sources of prebiotics molecules. As a result, various molecules have been suggested as candidate prebiotics or developed as commercially available functional foods. These include fructans (e.g., inulin, fructo-oligosaccharide), galacto-oligosaccharides, starch (e.g., resistant starch), glucose-derived oligosaccharides (e.g., polydextrose), other oligosaccharides (e.g., pectin), and non-carbohydrate oligosaccharides (e.g., flavanols) [3]. As each of the prebiotics has distinct properties (e.g., molecular structure and weight), their effects on human health could vary.

In general, the majority of prebiotics have been evaluated by in vivo experiments using mice models due to difficulties in in vivo studies of humans. However, as mice and humans have different physiologies, and as the gut microbial structures of mice are different from those of humans [7], the effects of prebiotics should not be translated for human use. On the other hand, the in vivo study of humans also has drawbacks due to difficulties in maintaining experimental conditions. Moreover, the human gut microbiome is strictly subject-dependent and can change dynamically depending on several factors, such as age, dietary habits, and medical history [8,9,10,11,12]. Therefore, there is a demand of the development of a new approach that allows evaluation of prebiotic potential for the human gut microbiome.

In this study, we digested several prebiotics using acids and gastrointestinal enzymes and fermented them with human fecal bacteria to simulate human consumption of prebiotics in vitro. We investigated the effects of those prebiotics on fecal microbial composition and bacterial SCFA production. We further implemented machine-learning models to predict the amount of SCFAs based on the fecal microbial composition in order to investigate whether this in vitro approach can be used to evaluate prebiotics potential. The present study provides a fundamental idea for the development of a fast and cost-effective approach that would make it possible to find more reliable prebiotics for human use.

## 2. Materials and Methods

### 2.1. In Vitro Gastrointestinal Digestion and Fecal Fermentation

In vitro gastrointestinal digestion (GID) and fecal fermentation of the eight different prebiotics were performed with fecal samples collected from 11 healthy adults (subject IDs: S1–S11), who had not taken any antibiotics for at least 6 months prior to the experiment. (+)-Arabinogalactan (ARA) was obtained from TCI (Tokyo, Japan) (CAS number 9036-66-2). Schizophyllan (SCH) was obtained from Quegen Biotech Co. Ltd. (Siheung, South Korea). Fructo-oligosaccharides (FOS) and inulin (INU) derived from chicory were obtained from Sigma-Aldrich (St. Louis, MO, USA) (CAS number F8052 and I2255 for each). Ginseng (Gins) was obtained from KGC (Daejeon, Korea) and laminarin (Lami) was obtained from Toronto Research Chemicals (Toronto, ON, Canada) (CAS No. 9008-22-4). Galacto-oligosaccharide (GOS) was obtained from CREMAR (Seoul, Korea) (CAS number 6587-31-1). D-(+)-Raffinose (RAF) was obtained from MB Cell (Seoul, Korea) (CAS number 512-68-6).

We followed the INFOGEST protocol [13] with several modifications for GID. In oral phase, 5 g (or 5 mL) of each prebiotic was suspended in 5 mL PBS and then mixed with 3.5 mL of simulated salivary fluid, 0.5 mL of 1500 U/mL α-amylase, 25 μL of 0.3 M CaCl_2_, and 975 μL of distilled water, making a total volume of 10 mL. The oral phase solution was placed on a rotary shaker (150 rpm) for 2 min at 37 °C. In the gastric phase, 10 mL of the oral phase solution was mixed with 7.5 mL of simulated gastric fluid, 1.6 mL of 25,000 U/mL porcine pepsin, and 5 μL of 0.3 M CaCl_2_. HCl was applied to adjust pH to approximately 2.5–3 and 695 μL of distilled water was subsequently, added making a total volume of 20 mL. The gastric phase digestion was performed at 100 rpm for 2 h. In the intestinal phase, 20 mL of the gastric phase solution was mixed with 11 mL of simulated intestinal fluid, 5 mL of 800 U/mL pancreatin, 2.5 mL of 160 mM bile salt, and 40 μL of 0.3 M CaCl_2_. NaOH was applied to adjust pH to 7 and distilled water was added, making a total volume of 40 mL. The intestinal phase digestion was performed at 100 rpm for 2 h. The final digested fluid was promptly placed inside liquid nitrogen, freeze-dried, and stored at −20 °C until fecal fermentation.

For fecal fermentation, fecal samples were collected in sterile, capped sampling cups, which were sealed immediately and transferred into an anaerobic chamber (90% N_2_, 5% H_2_, and 5% CO_2_) (Bactron II Anaerobic Chamber; Shel Lab, Cornelius, OR, USA). The fecal samples were suspended (20% *w*/*v*) in phosphate-buffered saline (PBS; 8.0 gL^−1^ NaCl, 0.2 gL^−1^ KCL, 1.15 gL^−1^ Na_2_HPO_4_, 0.2 gL^−1^ KH_2_PO_4_, 0.2 gL^−1^ L-cysteine hydrochloride), which had been placed inside the anaerobic chamber a day before fecal fermentation. The suspended samples were subsequently sieved through filters with pore sizes of 250 μm and 150 μm to remove fecal debris. For each subject, the fermentation of the blank control (no prebiotic) and each prebiotic was performed in triplicate. The experiments were performed in two batches on the same day: first five prebiotics (ARA, FOS, GOS, INU, RAF) and then the rest (SCH, Gins, and Lami). Feces of one subject (S4) ran out during the first batch, and the fecal fermentation with the stools of two subjects (S1 and S10) was performed on a different date. Therefore, there were no SCH, Gins, and Lami samples for S4 and there were two blanks for S1 and S10.

Fecal fermentation was performed as previously reported [14] with minor modifications. First, 800 μL of basal culturing medium and 100 μL of GID product (10%) were added to each well of a deep 96-well plate and 100 μL of fecal sample suspended in PBS (20%) was inoculated. The plate was covered with a silicon gel mat and the contents were incubated in digital shakers (MS3, IKA, Staufen, Germany) for 6 h at 37 °C at 500 rpm in an anaerobic chamber. Aliquots of the blank controls and samples were stored at −80 °C until DNA extraction and quantification of SCFAs.

Sample collection and the experimental procedures were approved by the Institutional Review Board (IRB) of Jeju National University (JJNU-IRB-2018-040-002).

### 2.2. Quantification of SCFAs

To extract SCFAs from the fecal microbiota, we followed the method described by Singh et al. (2021) [15] with slight modifications. The frozen fermented products were thawed on ice and 200 μL of each sample was added to 1 mL of absolute methanol. The mixture was vortexed for 2 min for homogenization and the pH of the mixture was adjusted to 2–3 using HCl. The mixture was incubated for 10 min at room temperature with repeated homogenization every 3 min and centrifuged at 15,000 rpm at 4 °C for 3–5 min until supernatant was observed to be transparent. The supernatant was collected in a 1 mL syringe and filtered with a membrane with pore size of 0.45 μm. The sample was stored in a liquid nitrogen tank until further processes.

For quantitative analysis of SCFAs (acetate, propionate, and butyrate), we adapted the method described by Scortichini et al. [16]. A gas chromatographic system (GC2010, Shimadzu, Japan) equipped with an auto injector (AOC-20i) and flame ionization detector (FID) was used with a nitroterephthalic acid-modified polyethylene glycol (PEG) column (20 m × 0.25 mm I.D., 0.25 μm film thickness; DB-FFAP, Agilent, Santa Clara, CA, USA). The inlet temperature was maintained at 230 °C and the injection was performed in splitless mode (1:10 ratio). The initial oven temperature was set to 80 °C for 3 min and incrementally increased to 200 °C with a rate of 15 °C/min. The temperature was held at 200 °C for 3 min and increased to 230 °C with a rate of 5 °C/min. The final hold was made at 230 °C for 10 min. Hydrogen was used as a carrier gas with a flow rate of 40 mL/min. The FID temperature was maintained at 280 °C. The SCFAs in each sample were identified by comparing retention time with reference standards. A stock solution of SCFAs was used as a standard to construct calibration curves for each compound.

### 2.3. Bioinformatics

Sequence reads were processed in Mothur v. 1.47.0 following the MiSeqSOP (https://mothur.org/wiki/miseq_sop/, accessed on 1 July 2022). Briefly, for quality control, sequence reads with ambiguous base pairs were removed and the cutoff length for homopolymers was set to 8 bp. Only the sequence reads within the length range of 350 bp to 550 bp were selected. Sequence reads were aligned against the Silva v. 138 database [17]. Chimeric sequences were removed through the VSEARCH algorithm [18]. Sequences were classified against RDP database v. 18 [19] and sequences classified as Chloroplast, Mitochondria, unknown, and Eukaryota were removed. Sequences with 97% similarity were clustered into a single operational taxonomic unity (OTU) using the Opticlust algorithm [20]. To predict functional profiles of microbial communities based on the 16S rRNA gene sequences, we used PICRUST2 v. 2.4.2 [21]. Sequence data normalized with 5047 reads per sample were used for PICRUST2 analysis.

### 2.4. Statistical Analysis

Differential abundance analysis was performed with the ALDEx2 package [22] in R to figure out the differentially abundant genera and metabolic pathways when compared to control samples (no prebiotics). Prior to alpha and beta diversity calculations, sequences were subsampled into 5047 reads per sample. The differences in Shannon diversity between each treatment were tested through an analysis of variance (ANOVA) test and pairwise comparison was performed with Tukey’s honestly significant difference (HSD) test. When the assumptions for the ANOVA test could not be met, a Kruskal–Wallis test was performed instead with Dunn’s test as a post hoc test. Principal coordinate analysis (PCoA) was performed with the R vegan package [23] to visualize the Bray–Curtis distance between each sample calculated with square root-transformed reads. Permutational multivariate analysis of variance (PERMANOVA) was performed with the adonis2 function in the R vegan package for each subject to test if there was a significant difference in different uses of prebiotics. A Mantel test based on Spearman’s rank correlation was performed with R to test whether initial microbial community structure was correlated with increased amounts of SCFAs. The within-group Bray–Curtis dissimilarity of the initial microbial community was averaged prior to the Mantel test. The increased amounts of SCFAs were averaged for replicates and the Euclidean distance was calculated as the input for the Mantel test.

### 2.5. Machine Learning

We adapted the method described in Zhou et al. [24] for machine learning. To predict the amount of SCFAs based on the microbial community data and on the predicted functional profiles, we applied seven different machine-learning algorithms: (1) random forest (RF) [25], (2) extreme gradient boosting (XGBoost) [26], (3) support vector machine (SVM) [27], (4) lasso [28], (5) ridge [29], (6) elastic net (ENet) [30] and (7) k-nearest neighbor (KNN) [31]. Machine learning was performed using the scikit-learn module [32] and the xgboost module in Python v. 3.9.7. The treatments/conditions were not considered as dependent variables in the models. Prior to building the models, genera/pathways with fewer than 10 reads were removed and the relative abundance of each genus/pathway in each sample was calculated. The relative abundances of genera/pathways were averaged for the three replicates in each treatment. In total, 98 cases were used for training and testing the models. The hyper-parameter of each model was tuned using the GridSearchCV class in the model_selection module of the Python scikit-learn package. Based on the best combination of hyper-parameters (Appendix A), we performed 100 rounds of fivefold cross-validation for each model algorithm, using different random splits for each round. The predicted amounts of SCFAs from each of the rounds were averaged to obtain the final predicted value. The importance of each genus/pathway in the random forest model was computed with the feature_importances function at each round and averaged to obtain the final value.

## 3. Results and Discussion

### 3.1. SCFA Production after Prebiotics Treatments

The amount of SCFAs produced in the gut is often used as an indicator of the health status of the gut, as they work as an important metabolite in the human body. The total amount of SCFAs generally increased when prebiotics were provided, except for SCH in all of the studied subjects (Appendix A). The increase was less prominent in the samples treated with ARA and Lami. Overall, the prebiotics tested in this study worked well, except for ARA, SCH, and Lami, which have branches and therefore more complexed molecular structures. For example, arabinogalactan consists of a galactan backbone with galactose and arabinose side chains, and both SCH and Lami are β-1,3 beta-glucans with β-1,6 branching, with the degree of branching being higher in the case of SCH [33]. In addition, the molecular weight of SCH is about 1.78 × 10^6^ Da) [34], which was higher than other prebiotics tested. On the other hand, these low productions of SCFAs could have been due to intrinsic limitations of the in vitro analysis, where all of the enzymatic reactions were performed in a shaker to mimic peristalsis. Further study is needed to investigate if extending time for GID and/or fecal fermentation could allow digestion of these high-molecular-weight and long-chained prebiotics in this in vitro environment.

### 3.2. SCFA Production and Fecal Microbiota of Each Subject

The initial microbial community was distinct in each of the subjects at the OTU level (Appendix A). Taxonomic composition analysis showed Firmicutes and Bacteroidetes were the two most abundant phyla in most of the subjects (Appendix A) and *Prevotella* was the most prevalent bacteria at the genus level, followed by *Phocaeicola* and *Bifidobacterium* (Appendix A). Subjects S1 and S6 had very small proportions of *Prevotella*. The microbiota of all of the studied subjects was divided into two groups: one was dominated by *Prevotella*, followed by *Faecalibacterium* and *Holdemanella*, and the other was dominated by *Phocaeicola*, followed by *Bifidobacterium* and *Blautia*, based on the centered log ratio value. Arumugam et al. (2011) [35] categorized gut microbes of humans into three different enterotypes which were identifiable by variations in the levels of *Bacteroides* (*Phocaeicola*), *Prevotella*, and *Ruminococcus*. These ”enterotype” differences expectedly showed variations in bacterial metabolic functions. We, however, did not observe a significant correlation between the increased amount of SCFAs after treatment of each prebiotic and these initial microbial community structures (Mantel static r = 0.138, *p* = 0.279). Our results showed that the increase in SCFA amounts was highly dependent on the type of prebiotics used rather than the initial microbiota of the subjects. The effective prebiotics (i.e., FOS, Gins, GOS, INU, and RAF) seemed to work well regardless of different enterotypes, at least in the present study. Fu et al. [36], however, reported that personal gut microbiota difference affects the effects of prebiotics. Therefore, a comprehensive study on more subjects with diverse enterotypes should be designed to test this further.

### 3.3. Microbiota Shifts after Prebiotics Treatment

The results in Figure 1A show that prebiotics treatment slightly shifted the microbial community. These shifts, however, were found to be significant when each subject’s fecal microbiota was separately plotted (Appendix A). PERMANOVA results showed significant shifts in microbiota caused by prebiotics treatment. In many cases, samples treated with Gins, GOS, and RAF clustered together, while samples treated with INU and FOS clustered together. The blank samples clustered with the samples treated with SCH, ARA, and Lami, suggesting the effects of these prebiotics on the microbiota were limited. To back this up, we observed significant associations between the extent of community shift and the increased amounts of short-chain fatty acids (Figure 1B). Our results showed that samples treated with prebiotics similar in their molecular structure clustered together. For example, samples treated with INU and FOS clustered together, as both INU and FOS are linear fructosyl polymers/oligomers linked by β-(2,1) bonds, attached to a terminal glucosyl residue by an α-(1,2) bond. Furthermore, samples treated with GOS, RAF, and Gins clustered together. GOS has the same composition as RAF, a trisaccharide composed of galactose, glucose, and fructose, but GOS has additional galactose molecules [37]. Ginseng is a multiplex of several different components, including carbohydrates (50–60%); N-containing substances (12–15%), such as proteins, peptides, and alkaloids; saponin (3–6%); and others [38]. There could be substances similar to GOS and RAF included in ginseng, resulting in similar microbial communities after treatment. In addition, the results in Figure 1C showed that Shannon diversity decreased after prebiotics treatment, implying a force of selection.

### 3.4. Effects of Prebiotics Treatments on the Abundance of Genera and Predicted Metabolic Activities

Differential abundance analysis results showed increases in Bifidobacterium, Agathobacter, Roseburia, Parabacteroides, Fusicatenibacter, Prevotella, Bacteroides, Faecalibacterium, Collinsella, and Catenibacterium after fermentation of prebiotics in most of the cases when compared to blanks (Figure 2A). These genera are mostly known producers of SCFAs, including Bifidobacterium, Roseburia, Bacteroides, and Faecalibacterium [39,40]. Streptococcus, Ruminococcus, and Romboutsia, on the other hand, were decreased after prebiotics treatments in many cases. Most of the Streptococcus spp. in the digestive tract are known to be commensal bacteria [41,42], except for lactic acid-producing Streptococcus thermophilus, which is abundant in milk products and used for yogurt production [43]. Ruminococcus play important roles in the digestion of cellulose and resistant starches, both in humans and ruminants [44,45]. As the prebiotics used in this study did not include cellulose or resistant starches, there might not have been enough energy for Ruminococcus to proliferate. Species belonging to the genus Romboutsia were first isolated in 2014 from a healthy rat [46] and have also been isolated from the human gut samples of a 63 year old man who suffered from severe anemia with melaena [47]. Although their metabolic capabilities are largely unknown, they have been suggested to be potentially harmful, as they were more abundant in patients with neurodevelopmental disorders compared to healthy subjects [48]. Phocaeicola showed an inconsistent pattern where its abundance increased in many cases when treated with ARA, Gins, and GOS. There were more cases that had decreased abundances of Phocaeicola when treated with FOS and Lami.

The results in Figure 2B show that prebiotics treatments increased some of the beneficial metabolic pathways, such as SCFA production and vitamin biosynthesis (Figure 2B). For example, ”pyruvate fermentation to butanoate”, menaquinol biosynthesis, and ubiquinol biosynthesis were enriched after prebiotics treatments. Menaquinone is vitamin K2, which plays an important role in hepatic coagulation [4], and ubiquinol is a reduced form of coenzyme Q10, which acts as electron carrier in cellular respiration and thus works as an antioxidant [49,50].

These results suggest that our in vitro GID followed by fecal fermentation successfully captured beneficial shifts both in microbiota and predicted metabolic activity profiles, although some prebiotics did not show beneficial effects. In this study, tested prebiotics were subjected to gastrointestinal digestion, although prebiotics are known to be resistant to mammalian digestive enzymes and gastric acids. The implementation of the complete digestive processes allowed us to mimic the human digestive system and confirm that the tested materials were still effective after the gastrointestinal digestions. Moreover, this system could be applied to evaluate the effects of non-prebiotic foods on gut microbiota [15].

### 3.5. Predicting Effects of Prebiotics with Machine-Learning Techniques

To further utilize the data collected in this study, we constructed machine-learning regression models that could predict the amount of SCFAs based on the microbial community composition of the studied samples. Overall, we were able to obtain well-performing models in which the correlation coefficients between the predicted amounts of short-chain fatty acid and the observed amounts of SCFAs were as high as 0.7 to 0.8 (Appendix A). The machine-learning models that predicted the amounts of SCFAs based on the metabolic pathways had correlation coefficients between the predicted and observed values as high as 0.6 to 0.7 (Appendix A), which is high but generally lower compared to the models constructed based on genus composition. It should be noted that the models based on pathways may suffer “double-speculations”, as the metabolic pathways had already been predicted once by PICRUST2 based on the community composition.

Feature importance in the random forest models calculated based on the mean squared errors indicated *Romboutsia* as the most important feature for predicting the amount of acetate, followed by *Dorea*, Clostridiales Incertae Sedis XIII unclassified, and *Bifidobacterium* (Figure 3A). The most important feature when predicting the amount of propionate was *Roseburia*, followed by *Megamonas*, *Parolsenella* and Clostridiales Incertae Sedis XIII unclassified. In the case of butyrate, *Faecalibacterium* was the most important feature, followed by *Prevotella*, *Roseburia* and Clostridiales Incertae Sedis XIII unclassified. Feature importance in the random forest models indicated catechol degradation II (meta-cleavage pathway) as the most important feature for predicting the amount of acetate, 1-4-dihydroxy-6-naphthoate biosynthesis II for predicting the amount of propionate, and L-histidine degradation I for predicting the amount of butyrate (Figure 3B).

The machine-learning models showed promising performance, suggesting the possible usage of microbiome data for the prediction of SCFA amounts. In this study, we were not able to construct a separate model for each of the prebiotics using only the initial gut microbiome as features because we had only 11 subjects. With increased numbers of subjects, it would be possible to construct a model to predict the best performing prebiotics for different individuals. Through such a model, we might be able to pick up personalized prebiotics that work best with individual gut microbiota.

In this study, we used the genus-level compositional data as the input for machine learning because OTU tables are sparse and include too many zeros, which could potentially bring bias/noise when constructing machine-learning models [24]. Future studies may apply long-read metagenomic or amplicon sequencing technology, such as SMRT and MinION, to develop more robust ML models with higher taxonomic resolution.

## 4. Conclusions

In this study, we were able to assess the performance of eight different prebiotics through an in vitro gastrointestinal digestion and fecal fermentation experiment. Prebiotics treatment elevated the amount of SCFAs and shifted the gut microbial communities, increasing the abundance of favorable bacterial groups with the ability to produce SCFAs. There was also an increase in beneficial metabolic activities after prebiotics treatment. The results from this study can be applied to probiotics studies and provide guidelines for in vivo experiments. The machine-learning approach could also be expanded to find personalized prebiotics treatments.

## Figures and Tables

**Figure 1 foods-11-02490-f001:**
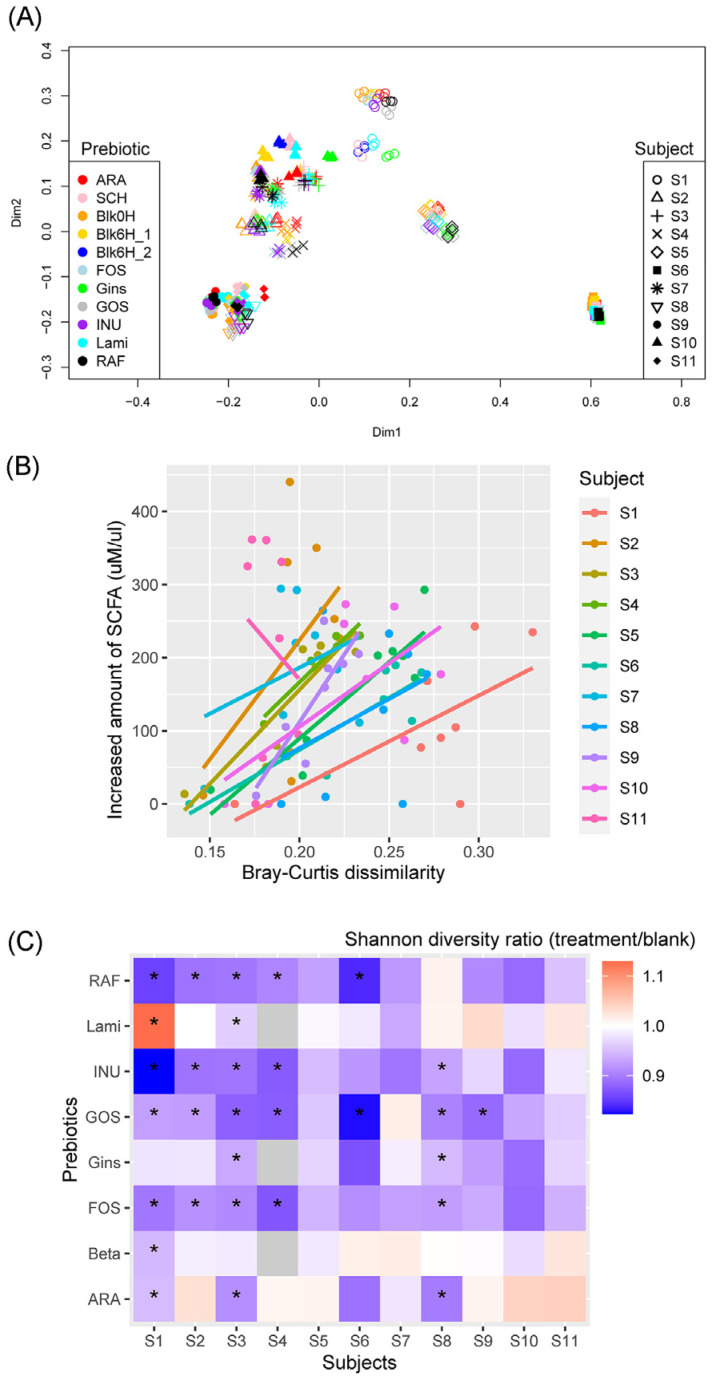
(**A**) Two-dimensional plot of the principle coordinate analysis (PCoA) based on Bray–Curtis distances of bacterial communities between samples. (**B**) A scatter plot with linear regression lines showing the relationship between community shifts and the increase in the SCFA amount. Bray–Curtis distances between treatment samples and blanks in each subject were calculated and plotted together with the increased amount of SCFAs. (**C**) A heatmap showing the ratio of Shannon diversity between treatment samples and corresponding blank samples. Asterisk (*) denotes significant difference in Shannon diversity between treatment samples and corresponding blank samples.

**Figure 2 foods-11-02490-f002:**
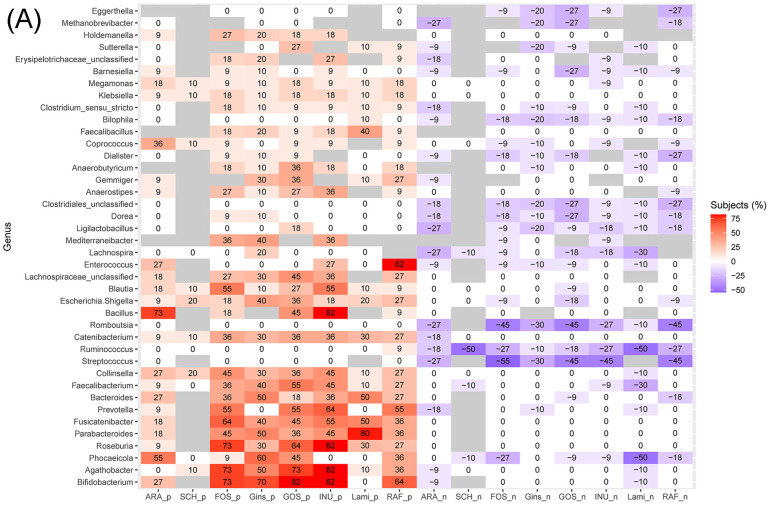
Heatmaps showing the percentages of subjects who had significantly increased (red) or decreased (blue) abundance of (**A**) genera and (**B**) metabolic pathways after prebiotics treatment when compared to blanks. Only the 40 genera (or pathways) that had the highest numbers of cases (subjects) showing significance in the differential abundance analysis are shown. To avoid noise, we used stricter criteria for pathways: only the cases that had centred log-ratio values between each treatment and the corresponding blank sample higher than 2 were included.

**Figure 3 foods-11-02490-f003:**
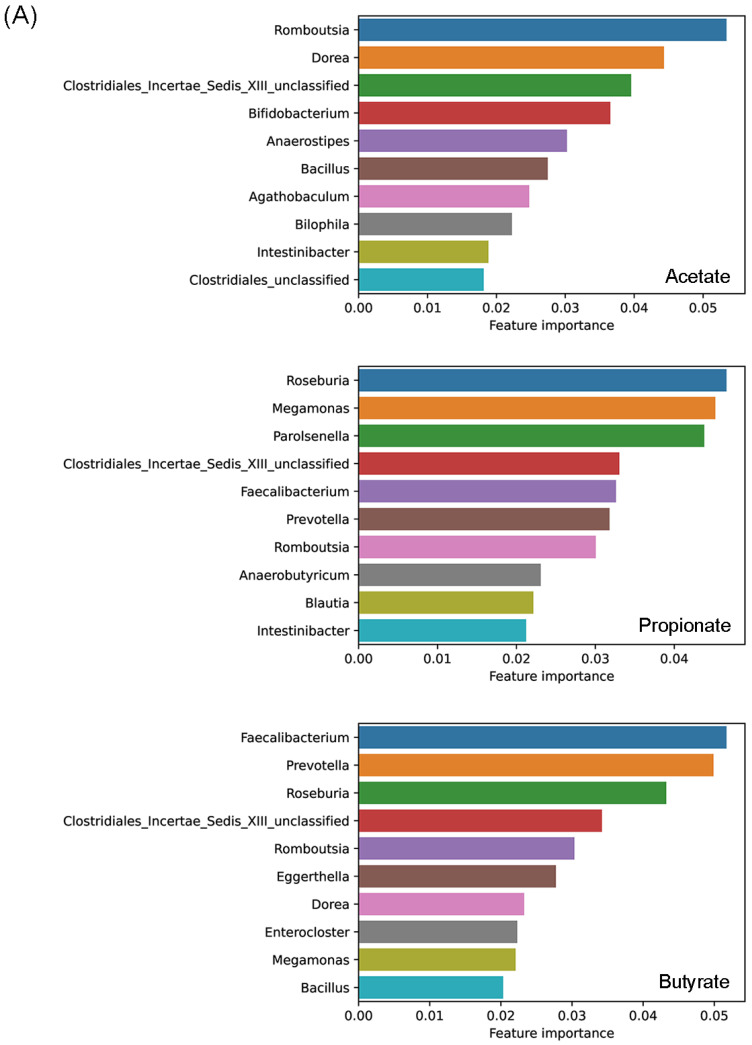
Important (**A**) genera and (**B**) pathways selected by feature importance in the random forest models calculated based on the mean squared errors.

## Data Availability

FASTQ Illumina sequence data have been deposited at the Sequence Read Archive (SRA) under project ID PRJNA847726.

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
