# Peer review of "Evaluation of Prebiotics through an In Vitro Gastrointestinal Digestion and Fecal Fermentation Experiment: Further Idea on the Implementation of Machine Learning Technique"

_foods, 2022, doi:10.3390/foods11162490_

Round 1

Reviewer 1 Report

In this study, Song et al. designed a study to investigate the evaluation of the production of SCAF of prebiotics using an in vitro fermentation experiment, and constructed a machine learning models to predict the amount of SCFA using gut microbiota profile. In general, this study is interesting, and the manuscript was well written. However, there are some concerns to be addressed.

confusing

Comments to the author:

Line 213: the authors divided the subjects into two groups, and found that the initial microbial community was not significantly correlated with the SCFA amount. These findings should be discussed.

The statement in Line 223 and line 17 seems to be contradictory.

Whether ML models based on species or higher taxonomic levels would be more robust. It should be discussed.

Author Response

Line 213: the authors divided the subjects into two groups and found that the initial microbial community was not significantly correlated with the SCFA amount. These findings should be discussed.

Thanks for your comment, and we now added more discussion on this.

Result and discussion, line 226-232:

Our results showed that the increase in SCFA amount was highly dependent on the type of prebiotics used rather than initial microbiota of the subjects. The effective prebiotics (i.e., FOS, Gins, GOS, INU, and RAF) seem to work well regardless of different enterotypes at least in the present study. Fu et al. [36], however, reported that personal gut microbiota difference affects effects of prebiotics. Therefore, a comprehensive study on more of the subjects with diverse enterotypes should be designed to test this further.

Reference:

36. Fu, T.; Zhou, L.; Fu, Z.; Zhang, B.; Li, Q.; Pan, L.; Zhou, C.; Zhao, Q.; Shang, Q.; Yu, G. Enterotype-Specific Effect of Human Gut Microbiota on the Fermentation of Marine Algae Oligosaccharides: A Preliminary Proof-of-Concept In Vitro Study. Polymers 2022, 14, doi:10.3390/polym14040770.

The statement in Line 223 and line 17 seems to be contradictory.

Thanks for your comments. By “Line 223” we meant to describe that prebiotics treatment slightly shifted microbial community, whereas “Line 17” describes how these shifts were correlated to the amount of SCFA increase. Therefore, we believe these two sentences are not contradictory.

Whether ML models based on species or higher taxonomic levels would be more robust. It should be discussed.

Thanks for pointing this out. We agree that many readers would wonder why we used the genus level for ML models. We now added discussion on this issue.

Result and discussion, line 327-331:

In this study, we used the genus-level compositional data as the input for machine learning because OTU tables are sparse and include too many zeros, which could potentially bring bias/noise when constructing machine learning models [24]. Future studies may apply long-read metagenomic or amplicon sequencing technology such as SMRT and MinION to develop more robust ML models with higher taxonomic resolution.

Reference:

24. Zhou, Y.-H.; Gallins, P. A Review and Tutorial of Machine Learning Methods for Microbiome Host Trait Prediction. 2019, 10, doi:10.3389/fgene.2019.00579.

Reviewer 2 Report

I consider that it is a good study, however, there are some considerations that must be settled in the introduction and in the writing of the conclusions:

 1. The prebiotics studied were subjected to a digestion process and it is assumed that these compounds have resistance to gastric acidity, hydrolysis by mammalian enzymes, and gastrointestinal absorption. It is important that somewhere in the introduction this statement is established.

2. What is the goal of having a complete digestive process. What changes are carried out by prebiotics in the upper gastrointestinal tract of prebiotics? Include bibliographic information about this in the introduction.

3. In the conclusions you conclude that "the results from this study can be applied to probiotics study and replace costly and laboriously animal-based in vivo experiments". I think this is not entirely true. These studies can give a guideline because there are many conditions that are not considered in these studies. For what I think is important, it is stated that they are guidelines that can be antecedents for in vivo studies, but they can never replace in vivo studies. Please reconsider that sentence.

Author Response

1. The prebiotics studied were subjected to a digestion process and it is assumed that these compounds have resistance to gastric acidity, hydrolysis by mammalian enzymes, and gastrointestinal absorption. It is important that somewhere in the introduction this statement is established.

We thank for your suggestions. We now added the followings in the introduction.

Introduction, line 27-32:

Prebiotics are defined as “a non-digestible food ingredient that beneficially affects the host by selectively stimulating the growth and/or activity of one or a limited number of bacteria in the colon, and thus improves host health” [1]. Therefore, prebiotics are resistant to gastric acid, hydrolysis by mammalian enzymes, and gastrointestinal absorption [2]. Prebiotics stay intact in the upper gastrointestinal tract, which is one of the important properties of prebiotics.

References:

1. Gibson, G.R.; Roberfroid, M.B. Dietary Modulation of the Human Colonic Microbiota: Introducing the Concept of Prebiotics. The Journal of Nutrition 1995, 125, 1401-1412, doi:10.1093/jn/125.6.1401.

2. Khangwal, I.; Shukla, P. Prospecting prebiotics, innovative evaluation methods, and their health applications: a review. 3 Biotech 2019, 9, 187-187, doi:10.1007/s13205-019-1716-6.

2. What is the goal of having a complete digestive process.

We performed a complete digestive process for better mimicry of in vivo experiments. This verifies that tested materials are still effective after going through gastrointestinal digestions. On the other hand, implementing GI digestive process allows us to evaluate effects of non-prebiotic foods on gut microbiota. We now added the following sentence to the discussion section

Results and Discussion, line 291-296:

In this study, tested prebiotics were subjected to gastrointestinal digestion although prebiotics are known to be resistant to mammalian digestive enzymes and gastric acids. The implementation of the complete digestive processes allows us to mimic human digestive system and confirm that tested materials are still effective after the gastrointestinal digestions. Moreover, this system could be applied to evaluate effects of non-prebiotic foods on gut microbiota [15].

Reference:

15. Singh, V.; Hwang, N.; Ko, G.; Tatsuya, U. Effects of digested Cheonggukjang on human microbiota assessed by in vitro fecal fermentation. Journal of Microbiology 2021, 59, 217-227, doi:10.1007/s12275-021-0525-x

What changes are carried out by prebiotics in the upper gastrointestinal tract of prebiotics? Include bibliographic information about this in the introduction.

Thanks for your comment. Please refer to our reply to the first comment.

3. In the conclusions you conclude that "the results from this study can be applied to probiotics study and replace costly and laboriously animal-based in vivo experiments". I think this is not entirely true. These studies can give a guideline because there are many conditions that are not considered in these studies. For what I think is important, it is stated that they are guidelines that can be antecedents for in vivo studies, but they can never replace in vivo studies. Please reconsider that sentence.

We agree with the reviewer and have revised the sentence as follows.

Conclusion, line 348-349:

The results from this study can be applied to probiotics study and can provide guidelines prior to in vivo experiments.